# Proteomics Analysis of R-Ras Deficiency in Oxygen Induced Retinopathy

**DOI:** 10.3390/ijms24097914

**Published:** 2023-04-26

**Authors:** Maria Vähätupa, Janika Nättinen, Ulla Aapola, Hannele Uusitalo-Järvinen, Hannu Uusitalo, Tero A. H. Järvinen

**Affiliations:** 1Faculty of Medicine and Health Technology, Tampere University, 33521 Tampere, Finland; 2Tampere University Hospital, 33520 Tampere, Finland

**Keywords:** R-Ras, oxygen-induced retinopathy, proteomics, angiogenesis, vascular permeability, leakage, hypoxia, blood-brain barrier, glycolysis, retina

## Abstract

Small GTPase R-Ras regulates vascular permeability in angiogenesis. In the eye, abnormal angiogenesis and hyperpermeability are the leading causes of vision loss in several ischemic retinal diseases such as proliferative diabetic retinopathy (PDR), retinal vein occlusion (RVO), and retinopathy of prematurity (ROP). Oxygen-induced retinopathy (OIR) is the most widely used experimental model for these ischemic retinopathies. To shed more light on how the R-Ras regulates vascular permeability in pathological angiogenesis, we performed a comprehensive (>2900 proteins) characterization of OIR in R-Ras knockout (KO) and wild-type (WT) mice by sequential window acquisition of all theoretical mass spectra (SWATH-MS) proteomics. OIR and age-matched normoxic control retinas were collected at P13, P17, and P42 from R-Ras KO and WT mice and were subjected to SWATH-MS and data analysis. The most significant difference between the R-Ras KO and WT retinas was an accumulation of plasma proteins. The pathological vascular hyperpermeability during OIR in the R-Ras KO retina took place very early, P13. This led to simultaneous hypoxic cell injury/death (ferroptosis), glycolytic metabolism as well compensatory mechanisms to counter the pathological leakage from angiogenic blood vessels in the OIR retina of R-Ras deficient mice.

## 1. Introduction

Retinal neovascularization and vascular hyperpermeability are the leading cause of blindness and visual impairment in several common ischemic retinal diseases, such as proliferative diabetic retinopathy (PDR), retinopathy of prematurity (ROP) and retinal vein occlusion (RVO) [1]. Retinal ischemia induces growth of neovessels, which are hyperpermeable and prone to bleeding into the retina and vitreous and ultimately causing loss of vision. Retinal neovascularization is also accompanied by fibrosis which may result in tractional retinal detachment, a dreaded and potentially blinding complication of ischemic retinopathy. Not only neovascularization but also hyperpermeability of retinal vasculature may result in vision loss. Retinal ischemia drives the production of cytokines, such as vascular endothelial growth factor-A (VEGF-A), which increase vascular permeability resulting in an accumulation of fluid in the macular edema) and subsequent vision loss [1].

VEGF-A, first discovered as a “vascular permeability factor”, a soluble protein secreted by tumors and shown to increase vascular permeability, is the most potent growth factor in inducing angiogenesis and vascular permeability [2]. Under physiologic conditions, vascular permeability is strictly controlled, but this control is lost in many diseases and the blood vessels become hyperpermeable, and they leak. In the retina, VEGF-A has a key role in inducing pathological angiogenesis and vascular leakage in retinopathies [1]. The introduction of anti-VEGF treatment has revolutionized the care of patients with retinal neovascular diseases and macular edema [1,3]. However, despite the progress of anti-VEGF therapy, there is a substantial proportion of patients with suboptimal or inadequate response, as well as unwanted side effects [4,5].

The major shortcoming of VEGF-inhibitor relates to their mechanism of action, the inhibition of angiogenesis. While VEGF inhibitors are highly potent in reducing macular edema and neovascularization, they do not relieve underlying hypoxia in the retina. The eradication of the neovessels by VEGF inhibitors may even worsen the underlying ischemia in the retina and subsequently drive the formation of new blood vessels by alternative molecular mechanisms in the hypoxic retina [4,6]. Thus, a better understanding of the factors controlling the vascular permeability in retinopathy is required to develop next-generation drugs for more efficient and selective treatment of retinopathy [4,5,7]. These drugs should be selective in their function; they should stabilize the pathological leaky blood vessels to ones that carry oxygen to address the hypoxia in the retina without having pathological permeability [4,5,7].

R-Ras gene is a small GTPase and a member of the Ras superfamily that contains numerous cancer-promoting genes, known as oncogenes [8,9]. However, the biological function of R-Ras is generally the opposite of its better-known family members. It maintains cellular quiescence and adherence and inhibits cellular proliferation, all biological effects that are generally the opposite to the other Ras family members’ functions, which play a prominent role in cancer formation and progression [8,9,10,11,12,13]. The biological function of R-Ras was revealed by the generation of R-Ras deficient mice, which revealed that the R-Ras regulates blood vessel maturation and vascular permeability in angiogenesis [12,14,15,16]. We have demonstrated that R-Ras is involved in the pathogenesis of retinopathies; its deficiency leads to hyperpermeability in sprouting angiogenesis in the experimental retinopathy model and the abundant R-Ras expression is lost from retinal blood vessels in human DR [17]. The immature leaky retinal blood vessels from human DR patients lack R-Ras expression completely and the retinal R-Ras expression correlates with vascular leakage in human DR: less R-Ras expression, more leakage in human DR [17].

Oxygen-induced retinopathy (OIR) is a widely used experimental disease model for ischemic retinopathies. It recapitulates key pathological features, the retinal neovascularization and the enhanced vascular permeability, of the human retinopathies [10,18,19,20,21,22,23]. It has been demonstrated that in addition to VEGF, other factors, such as erythropoietin, angiopoetin-2, and angiopoietin-like 4, participate in the regulation of vascular permeability in OIR [24]. Unfortunately, most efforts to control the process of angiogenesis and pathological vascular permeability have mainly focused on VEGF-A and its receptors, while other factors regulating this process have not received the scientific scrutiny they warrant to thoroughly understand the biological process [18,25]. R-Ras deficiency in the knockout (KO) mice leads to vascular hyperpermeability in OIR [17]. To further understand the pathological phenotype of R-Ras KO mice in OIR and its contribution to pathological leakage in OIR and human retinopathies, we have carried out a comprehensive proteomics analysis of normoxic and OIR retina in R-Ras KO and WT at various points along OIR using recently developed mass spectrometry technique, sequential window acquisition of all theoretical mass spectra (SWATH-MS).

## 2. Results

### 2.1. Protein Profiles Are Associated with the Developmental Stage of the Retina in R-Ras KO and WT in OIR

We have recently demonstrated that SWATH-MS is very sensitive in identifying protein expression changes that take place in angiogenic blood vessels in the OIR retina [26]. This prompted us to characterize the phenotype of R-Ras deficiency in OIR by proteomics. The retina samples from WT and R-Ras KO from both normal (control) and OIR mice were collected at the following time-points along OIR: P13, 24 h after the removal from the hyperoxia chamber to understand the immediate response to hypoxia, then at P17, when the angiogenesis peaks in OIR and finally at P42 to see whether potential differences in protein expression induced during OIR persist in the retina [27]. Furthermore, we also collected normal retinas (no induction of OIR) at P13, P17, and P42 from both WT and R-Ras KO mice as baseline control samples at each analysis time point. Altogether, 76 samples, 19 from each study group, were analyzed by proteomics. Before the proteomics analysis was performed, we confirmed the hyperpermeability of the angiogenic vasculature in the R-Ras KO OIR retina by staining histological samples with an anti-fibrinogen antibody. Fibrinogen is a commonly used marker of blood extravasation and the fibrinogen staining demonstrated that R-Ras KO mice had substantially more fibrinogen leakage than WT mice in the angiogenic phase of OIR (Figure 1).

After obtaining the proteomic data from all 76 samples, we evaluated the protein patterns and differences between time points and the genotypes. Principal component analysis (PCA) was performed from the complete data, and the associated plotting results based on the first two components, suggest that there is a clear division between different time points of both control and OIR retina samples in both R-Ras KO and WT samples (Figure 2). In addition to the changes related to different time points, some separation was seen between control and OIR retinas already at P13 and the OIR protein levels were substantially different from normal retinas at P17. The four groups were overlapping at P42. There was no longer detectable difference among normal/control retinal and OIR protein profiles in either R-Ras KO or WT samples. In summary, the developmental stage of the mouse retina appears to influence the protein expression levels in the retina most based on the first component and this difference is biologically more significant than changes induced by OIR or by R-Ras (WT vs. KO) genotype (Figure 2).

### 2.2. Vascular Leakage Is Elevated in the R-Ras KO Retinas after Return to Normoxia in OIR

Next, we performed a two-way ANOVA analysis to identify differences between the groups. Altogether 29 proteins at P13, 421 at P17 and 3 at P42 had ANOVA model *p*-values (adjusted) < 0.05. Of these proteins, 10, 8, and 1 proteins in time points p13, p17, and p42, respectively, had log2 fold changes > log2(1.5) and statistically significant *p*-values (<0.05) when R-Ras KO and WT OIR groups were compared with Tukey’s multiple comparisons of means (Figure 3, Figure 4 and Figure 5). Appendix A displays complete sets of statistical results.

Six proteins showed a statistically significant difference to all other treatment groups in R-Ras KO OIR retinas at P13. Three of those proteins were plasma proteins: alpha-2-HS-glycoprotein/Fetuin-A (Ahsg), serum albumin (Alb), and apolipoprotein A-I (Apoa1). These plasma proteins were all up-regulated in OIR KO retinas compared to all other groups. Another protein showing enhanced expression in R-Ras KO in response to hypoxia was glial fibrillary acidic protein (Gfap), a well-known marker of cell injury in the retina and brain [24]. Furthermore, glutamine synthetase (Glul) was expressed at a significantly higher level in R-Ras KO than in WT OIR retina at P13 (Figure 3).

The cytoplasmic NADPH phosphatase, Hddc3/Mesh1, that leads to NADPH dephosphorylation and induces subsequently ferroptosis, a form of cell death [28,29], was also upregulated in R-Ras KO and R-Ras KO OIR retinas at P13 (Figure 3). MESH1 expression remained elevated in R-Ras KO OIR also at P17 and was significantly higher when compared to WT OIR and to WT control but not to KO control. RNA binding motif protein 42 (Rbm42) expression was significantly reduced in R-Ras KO OIR (Appendix A).

### 2.3. Compensatory Mechanisms Are Activated to Establish Vascular Supply and Vascular Barrier Function in R-Ras KO at the Peak of Angiogenesis

To explore the function of R-Ras in retinopathies further, we next examined the potential changes in the gene expression between KO and WT at the peak of angiogenesis, P17. As stated earlier, the NADPH phosphatase Hddc3/Mesh1 expression remained at a significantly higher level in KO OIR than in WT samples (Figure 4). The enhanced expression of Glul at P13 indicated that glycolysis is the energy pathway activated in the R-Ras KO retina to establish vasculature in OIR. In line with Glul overexpression, Pgm1, phosphoglucomutase that catalyzes the bi-directional interconversion of glucose 1-phosphate (G1p) and glucose 6-phosphate (G6p), the first intermediate substrate in glycolysis. Phosphomannomutase 1 (Pmm1) was also expressed more in R-Ras KO than in the WT OIR retina (Figure 4). It has been demonstrated that the deficiency of phosphomannomutase-activity leads to impaired endothelial barrier integrity [30].

### 2.4. Retinal Protein Expression in R-Ras KO Returns to Resemble WT Retina in OIR at P42

The P42 analysis point is used in the OIR model to see whether the potential biological changes induced by OIR in the retina are permanent or temporary. *p*-value and fold-change filtering reduced the number of statistically significant results to one protein, Pgm1 at P42 R-Ras KO OIR retina. Interestingly, the Pgm1 gene has an even higher expression level in the normoxic R-Ras KO retina than in the R-Ras WT OIR retina at P42 indicating that the glycolytic energy metabolism is the preferred way of producing energy when R-Ras is lacking (Figure 5).

**Figure 5 ijms-24-07914-f005:**
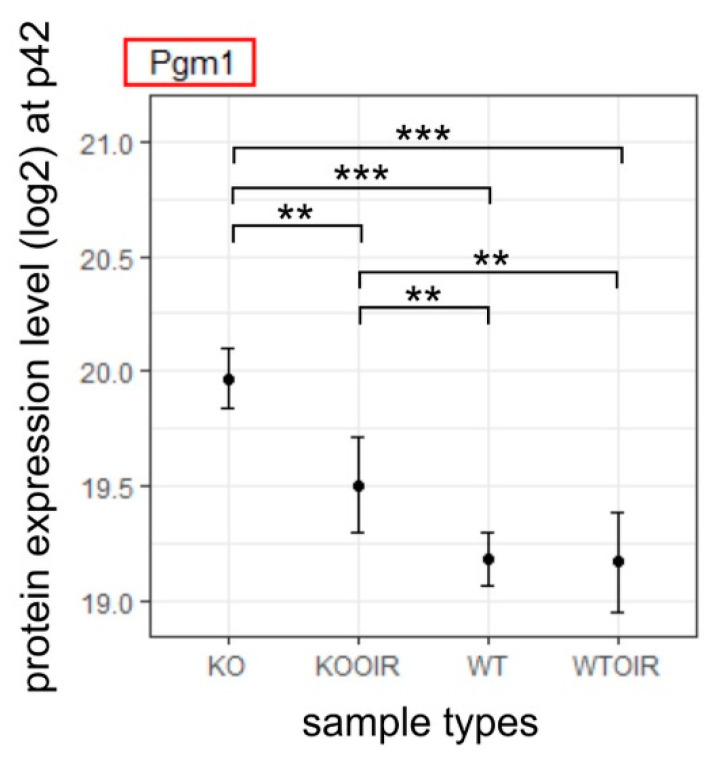
Differentially expressed protein between R-Ras KO OIR and WT OIR mice after OIR at P42. R-Ras KO and WT mice pups were exposed to hyperoxia-induced OIR, and the mice were returned to normal room air at P12. The retinas were harvested at P42, i.e., 30 days after removal from the hyperoxia chamber. All retina samples were analyzed by MS-SWATH and the data underwent statistical analysis to compare R-Ras KO and WT control and OIR samples at P42. The means and standard deviation bars are shown for all four groups, R-Ras KO and WT control and OIR samples for chosen protein at P42. The data were analyzed with ANOVA and Tukey multiple comparisons of means. **, *p* < 0.01; ***, *p* < 0.001. A recent study implicated Pgm1 function in long-term adaptation to hypoxia [31].

## 3. Discussion

The R-Ras deficient mice revealed that the primary function of R-Ras is blood vessel maturation and stabilization not only in pathological angiogenesis but also in physiological angiogenesis, i.e., in tissue repair [12,14,15,16,17,32,33]. We demonstrate in this study that the R-Ras KO mice have an increased vascular permeability (leaky) phenotype in the OIR and this leads to cellular injury/death as well as the induction of compensatory mechanisms to counter the deleterious vascular leakage from the vasculature.

R-Ras has a key role in controlling vascular permeability in angiogenesis [12,14,15,16,17,32,33] and has been recently implicated to play a prominent role in the progression of human neovascular retinal diseases [17], but also in peripheral arterial disease [34]. The pathological, immature neovessels lack R-Ras expression in the retina of the DR patients [17]. The R-Ras expression correlates inversely with leakage in DR; the more leakage there is in human DR, the less R-Ras is expressed in these blood vessels [17]. The increased R-Ras palmitoylation (inactivation) is another mechanism by which R-Ras can be inactivated in vascular diseases [34]. The most striking difference between the WT and R-Ras KO OIR retinas was the accumulation of plasma proteins in KO OIR already at P13 retinas as a sign of increased vascular permeability immediately upon the return to hypoxia.

Another important aspect arising from our study was the pronounced induction of Gfap in R-Ras KO mice during the hypoxic phase of OIR. Gfap is expressed in injured retinal Müller glial cells in different pathological conditions such as ischemia, trauma, retinal degeneration, and glaucoma [24,35,36]. Furthermore, its expression has been demonstrated in Müller glial cells in the hypoxic inner retina in OIR at P13 when these cells are damaged in the OIR model [24]. Simultaneously we also identified the enhanced expression Hddc3/Mesh1 in the R-Ras KO OIR retina both at P13 and P17. Hddc3/Mesh1 is the first identified cytoplasmic nicotinamide adenine dinucleotide phosphate (NADPH) phosphatase, that leads to NADPH dephosphorylation and induces subsequently ferroptosis [28,29], but is also indispensable for sustaining anaerobic metabolism [37]. Ferroptosis is a form of iron-dependent, programmed cell death. It is characterized by the accumulation of lipid peroxides and is distinct from other forms of programmed cell death such as apoptosis. Ferroptosis is a major contributor to cell death in ischemia–reperfusion injuries, such as brain stroke, ischaemic heart disease, and also retinopathies, due to oxidative stress induced by hypoxia [38,39,40]. Taken together with the substantially higher induction of Gfap and Hddc3/mesh1 in R-Ras KO retina in OIR, our results suggest that R-Ras is a vital protein for cells to withstand hypoxic tissue injury. Accordingly, R-Ras is also needed for the proper lumenization of angiogenic blood vessels in hypoxia [33], which is crucial to obtain proper perfusion in hypoxic tissue and to address the prevailing hypoxia. Furthermore, the R-Ras gene therapy also addresses hypoxia by inhibiting endothelial cell death/apoptosis [15]. It has been demonstrated in the eye that a selective deficiency of R-Ras in pericytes is enough to cause such a severe defect in retinal vascular supply that it leads to the development of microphthalmia, a disorder where the eye is abnormally small due to poor vascular supply [41].

Genes showing increased expression in R-Ras KO mice during the hypoxic phase of OIR were glutamine synthetase (Glul, GS) and phospoglycomutase-1 (Pgm1). Glutamine is a carbon and nitrogen donor for the production of different biomolecules and is involved in redox homeostasis. Most cells are capable of taking up glutamine, and therefore do not have a need to synthesize it. Certain cells express glutamine synthetase encoded by the gene Glul, which is an enzyme that can do de novo glutamine production from glutamate and ammonia [42]. Glutamate is a neurotransmitter that has excitatory effects on nerve cells, and that it can excite cells to their own death in a process called “excitotoxicity”. By continuously removing glutamate and ammonia from the extracellular fluid, Glul serves a detoxifying function. More recently, Glul has been demonstrated to have another important biochemical function. Namely, endothelial cells use glycolysis as a major source to fulfill their energy demands [42,43]. For that, Glul has been demonstrated as a vital enzyme for sprouting angiogenesis [42,44], while inhibition of Glul function leads to blood vessel pruning and normalization [45,46]. In line with the overexpression of Glul and Hddc3/Mesh1, Pgm1, which generates the first glycolytic intermediate after glucose (G6p), was significantly up-regulated in R-Ras KO OIR retina. As a matter of fact, Pgm1 was the only protein that was significantly overexpressed at P42 in R-Ras KO OIR compared to WT OIR, which is in line with a recent demonstration that the enhanced expression of Pgm1 is a sign of long-term adaptation to hypoxia [31]. Simultaneously with these protein changes, Rbm42 expression is reduced in R-Ras KO OIR. Rbm42 is an RNA-binding protein that regulates the pre-mRNA splicing of genes involved in growth [47]. It is also crucial for the maintenance of cellular ATP levels under stress [48] by controlling the translation of electron transport chain (ETC) genes through a repressive function [49,50]. Our results may indicate that the decreased expression of Rbm42 in R-Ras deficient OIR retina may result in increased mitochondrial ATP production by providing enough ETC enzymes for efficient glycolysis. The enhanced glycolysis has important implications for retinopathy. Namely, glycolysis-generated lactate is the major energy source for endothelial cells in OIR [51], but also for neurons and macrophages [52,53]. It is also crucial for maintaining proper pericyte function and the integrity of the blood-brain barrier [54]. Our results imply that R-Ras deficiency in angiogenesis is compensated by enhanced glycolysis in the hypoxic retina and the need for glycolysis to generate energy persists for a substantially long period of time in OIR retina to maintain some blood-retina integrity lacking in R-Ras deficiency.

Our current study demonstrates the hyperpermeability/increased vascular leakage in the R-Ras KO OIR retina. Phosphomannomutase 1 (Pmm1) expression was enhanced in R-Ras KO OIR. Pmm1 enzymatic activity is N-glycosylation (Pgm1 also has this enzymatic activity) that is crucial for maintaining endothelial barrier function in the blood-brain barrier [30,55,56] as well as for rescuing glycolysis in the ischemic brain [56]. Furthermore, PMM activity is required to maintain the function of the cell adhesion molecule, N-cadherin [57] which is crucial for pericyte-endothelial cell interaction in blood vessels. Interestingly, R-Ras signaling is also required for N-cadherin expression between pericytes and endothelial cells in angiogenic blood vessels [58]. Pmm1 and Pgm1 expressions might be a compensatory mechanism to counter the deleterious hyperpermeability in R-Ras deficiency during OIR.

Here we have used a proteomics analysis to understand how R-Ras regulates vascular angiogenesis and vascular permeability in OIR. Our results confirm the vascular hyperpermeability in R-Ras deficiency and indicate that the loss of R-Ras leads to severe retinal injury/cell death in OIR. We identify not only the pathological consequences of R-Ras deficiency in the retina but also the potential compensatory mechanisms in the retina to obtain protection from the pathological leakage during OIR-related angiogenesis.

## 4. Materials and Methods

### 4.1. M Mice and Mouse OIR Model

WT and R-Ras KO C57BL/6 mice were used for the study. The generation of R-Ras KO mice has been described in detail previously [15]. Before any experiments, R-Ras heterozygous mice were backcrossed eight times with the C57BL/6 strain to obtain homozygous WT and KO mice in the same genetic background. The mice were bred, and the genotype was determined by PCR in each animal as described previously [59]. Mice were housed under standard conditions with a 12-h dark/12-h light cycle and fed with standard laboratory pellets and water ad libitum.

The OIR model was performed as described in detail previously [19,20,22,60]. Briefly, to induce retinopathy in the retina, the pups and their nursing mothers were exposed to 75% oxygen in a custom-made chamber (ProOx P110 oxygen controller; Biospherix Ltd., Parish, NY, USA) at postnatal day 7 (P7) for 5 days until P12 when they were returned to normal room air. Mice were sacrificed and retinas collected at P13 (early hypoxic phase), at P17 (late hypoxic phase and the peak of neovascularization), and at P42 (after vascular recovery) to assess the effect of OIR on the retinal proteome. Control animals were housed under normal room air conditions and retinas were harvested on corresponding days. The study design as well as the number of retinas harvested for analysis at each time-point are illustrated in Figure 6.

As postnatal weight gain has been shown to affect the revascularization rate of the retina in the OIR [20], only the pups weighing between 6.3 and 7.5 g at P17 were included in the study.

For the proteomic analysis, eyes/eyeballs were harvested into cold, sterile PBS. The retinas were dissected under the dissection microscope immediately. The retinas were then snap-frozen with liquid nitrogen and stored at −70 °C until sample preparation.

All animal experiments were conducted under ARVO Statement for the Use of Animals in Ophthalmic and Vision Research guidelines in accordance with protocols approved by the National Animal Ethics Committee of Finland (protocol # ESAVI/92/04.10.07/2014 and ESAVI/6421/04.10.07/2017).

### 4.2. Proteomics

#### Sample Processing and Proteomic Analysis

SWATH-MS proteomics was carried out essentially as described previously [26]. Both retinas from each animal were combined as a single sample and processed together for proteomics. Th samples were collected in the following fashion:

Briefly, proteins were extracted and homogenized from retinal tissues in RIPA cell lysis buffer. Total protein concentrations were measured using a Bio-Rad DC protein quantification kit (Bio-Rad, Hercules, CA, USA), and 50 µg of protein from each individual sample was prepared for MS analysis. An equal amount (four micrograms) of each sample was analyzed with NanoLC-TripleTOF (Sciex 5600+) mass spectrometry using the SWATH acquisition method. Two replicate MS analyses were performed for each sample. A detailed protocol of sample preparation and SWATH-MS analysis has been described in detail [26].

A relative protein library was created using Protein Pilot 4.7 (Sciex, Redwood City, CA, USA) and mouse UniprotKB/SwissProt protein database. In the library generation, a false discovery rate (FDR) of 1% was implemented. Quantification was performed by Peak Viewer and Marker Viewer programs (Sciex, Redwood City, CA, USA) as described previously [26].

### 4.3. Immunohistochemistry

For immunohistochemistry, the eyeballs were harvested, fixed immediately with 4% filter-sterile PFA for 4 h, and processed for paraffin embedding [26]. Five-micrometer thick sections were subjected to antigen retrieval (Tris-EDTA, pH9), blocked, and incubated with polycloncal anti-fibrinogen antibody (ThermoFisher Scientific, Waltham, MA, USA) followed by horseradish peroxidase (HRP) conjugated secondary antibodies.

### 4.4. Statistical Analysis

Overall, 2944 proteins were quantified from 76 samples (2 replicate MS runs each). The number of samples in each group is displayed in Figure 6. Log2-transformation was applied to the data and means of replicate MS runs were calculated for each protein and these values were used in further analyses.

Principal component analysis (PCA) was used to cluster the samples based on full proteomic profiles. Analysis of variance (ANOVA) was used to evaluate differences between the four sample groups, WT, KO, WTOIR, and KOOIR, at separate time points. Tukey multiple comparisons of means were used as a post-hoc test to examine the differences between means in different groups.

Benjamini–Hochberg procedure was used to correct for multiple testing. The significance level was set at *p* = 0.05 unless otherwise stated. All statistical analyses for the proteomics data were performed using R software version 4.1.2 (R Foundation for Statistical Computing, Vienna, Austria).

## Figures and Tables

**Figure 1 ijms-24-07914-f001:**
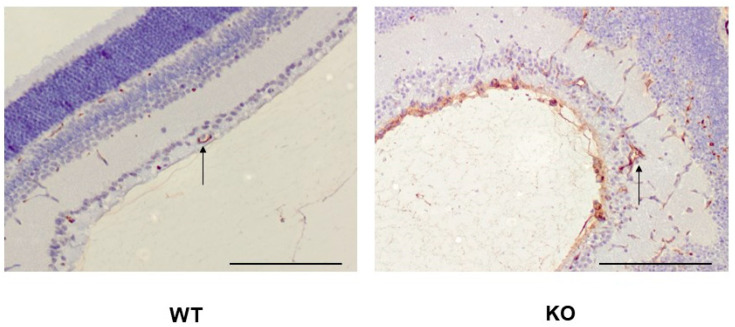
Vascular hyperpermeability in R-Ras deficient mice in OIR. WT and R-Ras KO OIR retinas were processed for immunohistochemistry and stained for fibrinogen as described in the Section 4. Substantially more abundant fibrinogen expression is seen in R-Ras KO than in WT mice not only in the retina, but also in the vitreous during the angiogenic phase of OIR (P17). Arrows represent blood vessels in the retina. Scale bar 150 µm.

**Figure 2 ijms-24-07914-f002:**
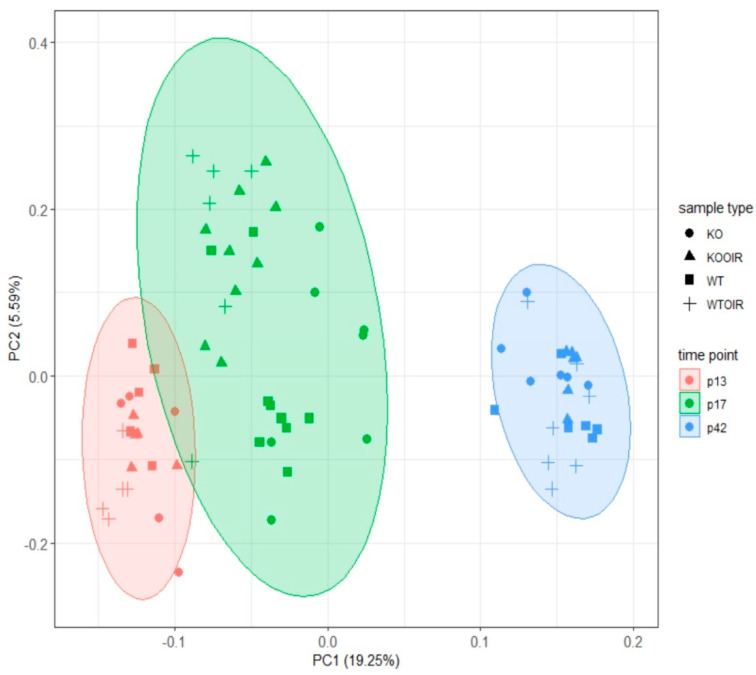
Principal component analysis (PCA) shows different clustering of retinal proteins according to the developmental stage of the retina both in R-Ras knockout (KO) and wildtype (WT) mice. The protein profiles obtained from control and oxygen-induced retinopathy (OIR) retinas are distinct from each other at P13 (red), P17 (green), and P42 (blue). The difference is explained by the developmental stage of the retina, not by the R-Ras genotype (KO vs. WT). Control (●/■) and OIR (▲/+) retina samples are also separated quite distinctively within the clusters at P13 and P17, but no more at P42.

**Figure 3 ijms-24-07914-f003:**
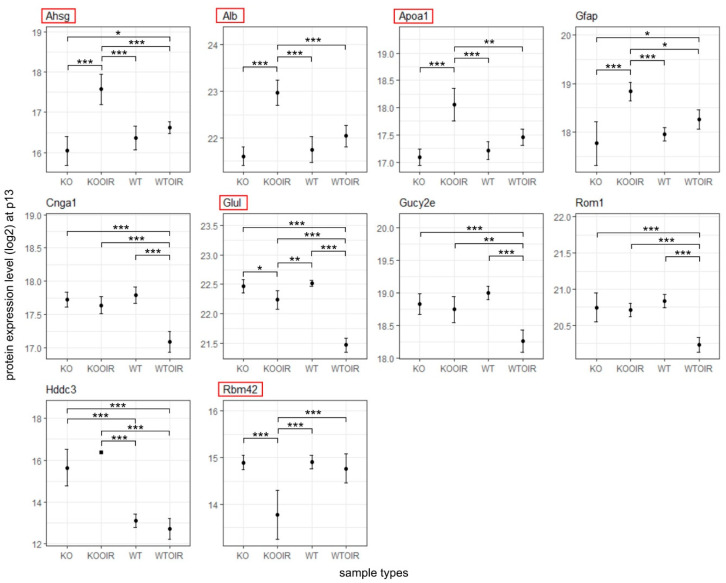
Differentially expressed proteins between R-Ras knockout (KO) and wild-type (WT) mice in oxygen-induced retinopathy (OI) during hypoxia (P13). R-Ras KO and WT mice pups were exposed to hyperoxia-induced OIR, and retinas were harvested at P13. The means and standard deviation bars are shown for all four groups, R-Ras KO and WT control, and OIR samples for chosen proteins during hypoxia at P13. The data were analyzed with ANOVA and Tukey multiple comparisons of means. *, *p* < 0.05; **, *p* < 0.01; ***, *p* < 0.001. The proteins that have been demonstrated to regulate vascular permeability are marked by red squares.

**Figure 4 ijms-24-07914-f004:**
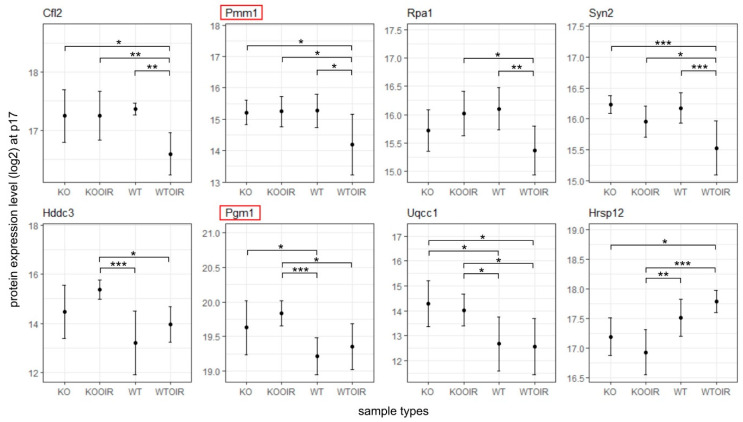
Differentially expressed proteins between R-Ras knockout (KO) and wild-type (WT) mice in oxygen-induced retinopathy (OIR) during hypoxia (P17). R-Ras KO and WT mice pups were exposed to hyperoxia-induced OIR, and retinas were harvested at P17 when angiogenesis peaks in OIR. The means and standard deviation bars are shown for all four groups, R-Ras KO and WT control and OIR samples for chosen proteins at the peak of neovascularization at P17. The data were analyzed with ANOVA and Tukey multiple comparisons of means. *, *p* < 0.05; **, *p* < 0.01; ***, *p* < 0.001. The proteins that have been demonstrated to regulate vascular permeability are marked by red squares.

**Figure 6 ijms-24-07914-f006:**
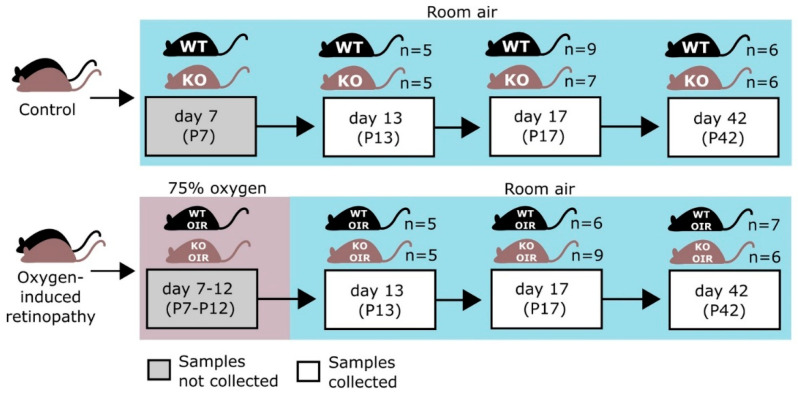
Outline of the study. Both R-Ras knockout (KO) and wild-type (WT) mice were exposed to 75% oxygen for 5 days from P7 to P12. Oxygen-induced retinopathy (OIR) retinas and retinas from the age-matched WT and R-Ras KO control mice were collected for proteomics analysis. The figure is adapted from Vähätupa et al., 2018 [26].

## Data Availability

Data presented in this study are available on request from the corresponding author.

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
