# Peer review of "Proteomics Analysis of R-Ras Deficiency in Oxygen Induced Retinopathy"

_ijms, 2023, doi:10.3390/ijms24097914_

Round 1

Reviewer 1 Report

This paper demonstrates the hyperpermeability/increased vascular leakage in R-Ras KO OIR retina, caused by PMM1 expression. It is one of the keys for PDR, ROP and RVO therapeutic target.

 Minor comments

Please put the scale in Figure 2 both photographs. Were these photographs collected at P13, P17, or P42? Add in the legend.

 Please clear Reference No.57 citation.

Author Response

Reviewer #1 (Remarks to the Author):

This paper demonstrates the hyperpermeability/increased vascular leakage in R-Ras KO OIR retina, caused by PMM1 expression. It is one of the keys for PDR, ROP and RVO therapeutic target.

Minor comments

Please put the scale in Figure 2 both photographs. Were these photographs collected at P13, P17, or P42? Add in the legend.

Reply: We thank reviewer # 1 for pointing out these omissions and we have revised the legend for Figure 2 accordingly. We have provided the scale bar as well as indicated the time-point when the samples were collected from OIR retinas (P17).

 Please clear Reference No.57 citation.

Reply: We thank reviewer # 2 for recognizing the omission of reference # 57 from the text. We have placed the reference # 57 to the right point in the text.

Reviewer 2 Report

The article “Proteomics analysis of R-Ras deficiency in oxygen induced retinopathy” demonstrates the role of R-Ras in regulating vascular permeability in pathological angiogenesis using OIR model. The article is well written, and results are very straight forward.

Here are some questions for authors:

1.     In Fig.2. scale bar is missing. What postnatal (P) time point the images were taken?

Both images look from different resolution, comment!

2.     It would be better if authors could show all the proteins that go up or down in one figure (such as bar graph) for all the groups at different time points (P13, P17 and P42).

3.     It would be beneficial for readers if authors could separate all the proteins that are significantly involved in the vascular leakage in separate graph (compared in all the groups).

4.     Did authors check the protein expression of either of these proteins among various groups after the proteomic analysis?

5.     Can authors comment more on the “Retinal protein expression in R-Ras KO returns to resemble WT retina in OIR at P42” that means Pgm1 protein involvement in vascular permeability is dependent on developmental stages or regression of neovascularization? What happened to other proteins that go up in P13 or P17 at time point P42?

6.     What are the authors comment on the regression of neovascularization in KOOIR mice as compared to WTOIR? Is that the same or any change?  

Author Response

Reviewer # 2

The article “Proteomics analysis of R-Ras deficiency in oxygen induced retinopathy” demonstrates the role of R-Ras in regulating vascular permeability in pathological angiogenesis using OIR model. The article is well written, and results are very straight forward.

Reply: We thank reviewer # 2 for generally positive comments on our manuscript.

Here are some questions for authors:

  1. In Fig.2. scale bar is missing. What postnatal (P) time point the images were taken?

Both images look from different resolution, comment!

Reply: We thank reviewer # 2 for pointing out these deficits in Figure 2. For unknown reasons the images taken form wild-type (WT) and knockout (KO) were indeed presented at different magnifications in the submitted manuscript. We retrieved the original images. The images are now presented in the same magnification in the revised version of the manuscript. We have provided the scale bar as well as indicated the time-point when the samples were collected from OIR retinas (P17) in the legend.

  1. It would be better if authors could show all the proteins that go up or down in one figure (such as bar graph) for all the groups at different time points (P13, P17 and P42).

      Reply: We want to remind that multiple statistical comparisons between study groups were performed at each time-point. These analyses yielded more than 200 proteins that demonstrated significantly different expression levels in some comparisons. Thus, we cannot present them all in the figures and we have decided to present only those that show statistically significant difference in all comparisons between the study groups. The rest of the data is presented in Suppl. Table 1.

  1. It would be beneficial for readers if authors could separate all the proteins that are significantly involved in the vascular leakage in separate graph (compared in all the groups).

Reply: We have revised the Figures 4 - 6 according to the suggestion provided above. The proteins involved in the regulation of vascular permeability have been identified by red squares in Figures 4 – 6. The legends for those figures in question have been revised accordingly.

  1. Did authors check the protein expression of either of these proteins among various groups after the proteomic analysis?

Reply: Our previous proteomics characterization of OIR model demonstrated that the changes in the protein expression identified in the proteomics analysis were confirmed both by the immunohistochemical examination and western blotting [28]. Thus, our focus was to carry out proteomics analysis with as high sample number as possible to guarantee the reproducibility. Altogether 152 retinas were subjected to proteomics analysis in the current study for that reason.

  1. Can authors comment more on the “Retinal protein expression in R-Ras KO returns to resemble WT retina in OIR at P42” that means Pgm1 protein involvement in vascular permeability is dependent on developmental stages or regression of neovascularization? What happened to other proteins that go up in P13 or P17 at time point P42?

Reply: The large majority of differences in the retinal protein expression induced either by R-Ras genotype (WT vs. KO) or by OIR itself at P13 and P17 disappear by P42 as the Pgm1 is the only protein showing statistically significant difference in all group-wise comparisons at that stage. No statistically significant differences were detected for rest of the proteins at P42. We believe that R-Ras KO retina is dependent on Pgm1 function as its enhanced expression persists in retina at P42.

  1. What are the authors comment on the regression of neovascularization in KOOIR mice as compared to WTOIR? Is that the same or any change?

Reply: The phenotype of R-Ras KO in OIR has been described in detail in our previous study [17]. The phenotype is unique; R-Ras KO and WT mice have similar angiogenic response in terms of re-vascularization and neovascularization, but despite this the vascular permeability is significantly enhanced in R-Ras KO mice in OIR [17]. We are not aware of any other genetic background that yields similar phenotype in OIR. This phenotype highlights the importance of R-Ras as a master regulator of vascular permeability in angiogenesis.

Round 2

Reviewer 2 Report

The article “Proteomics analysis of R-Ras deficiency in oxygen induced retinopathy” has explained all the questions raised.

But still one comment is not addressed properly.

In Fig 2. Still the images from WT and KO look from different resolution in the revised manuscript. It seems the authors kept scale bar on the same image from old version.

Check and revise.

Author Response

We thank reviewer # 2 for his/her comments. We want to raise few points before addressing the concern raised by the reviewer. The images presented in Fig. # 2 are taken from OIR at P17. The phenotype of R-Ras KO is vascular hyperpermeability in OIR, i.e. leakage, and thus the inner retina appears edematous in R-Ras KO OIR at P17. The 2nd issue related to the histological appearance of retina is the concave structure of eye ball. So the histological sections are never identical in retinal thichness. The aim of the Fig. # 2 is to present the blood vessels in the retina and the representative areas have been chosen based on the appearance of the blood vessels, not the thickness of retina.

We understand the point that the retinal thicknesses appear not to be identical between WT and KO mice and have made a small adjustment in that respect. However, we also encourage the reviewer # 2 to look at individual cells and structures such as blood vessels that can be easily identified and compare them between the presented images. We strongly believe that we are presenting images in identical magnification.

Concerning the scale bar, we venture to remind reviewer # 2 that the scale bar was actually missing from Fig. # 2 in the original submission and the scale bar was added to the revised version of the manuscript, thanks to the excellent review work done by both reviewers.